

# Intensity and dynamics of extreme cold spells of the 21st century in France from CMIP6 data

Camille Cadiou[1] and Pascal Yiou[1]

[1]Laboratoire des Sciences du Climat et de l'Environnement, UMR 8212 CEA-CNRS-UVSQ, IPSL and U Paris Saclay, 91191 Gif-sur-Yvette CEDEX, France

**Correspondence:** Camille Cadiou (camille.cadiou@lsce.ipsl.fr)

**Abstract.** Cold extremes significantly impact society, causing excess mortality, strain on healthcare systems, and increased demand on the energy system. With global warming, these extremes are expected to decrease, as observed in various indicators. This study simulates extreme cold spells of 15 days using a Stochastic Weather Generator (SWG) based on circulation analogues and importance sampling, adapted for CMIP6 data. Our results show that the most extreme cold spells decrease in intensity with global warming, making 20th-century-like events nearly impossible by the end of the 21st century. However, some events of late 20th century (e.g. 1985 in France) may still occur in the near future. Such events are associated with patterns of atmospheric dynamics that convey cold air from high latitudes into Europe. Those atmospheric circulation patterns show a consistent high-pressure system over Iceland and a strong low-pressure system over southwestern Europe in ERA5 and CMIP6 models. We show that nudging the SWG towards this type of pattern triggers extreme cold spells, even in a warmer world. We also evaluate the ability of CMIP6 models to represent such an atmospheric pattern. This study highlights the importance of understanding cold spell dynamics and the relevance of rare events algorithms and large ensemble models to simulate low-probability, high-impact events, offering insights into the future evolution of cold extremes.

## 1 Introduction

Cold extremes have significant impacts on society, with cold weather being linked to excess mortality and morbidity (Conlon et al., 2011; Masselot et al., 2023). Although mortality from heat extremes is increasing with global warming, more deaths are still attributable to cold extremes (Gasparrini et al., 2015), and healthcare systems in the mid-latitudes experience higher pressure during winter (Charlton-Perez et al., 2019).

Cold spells also have significant impacts on energy systems, affecting both energy demand and power supply as investigated in numerous studies (Añel et al., 2017; Bessec and Fouquau, 2008; Bloomfield et al., 2016; Davies, 1959; Jacob et al., 2018; Panteli and Mancarella, 2015; Pardo et al., 2002; Sailor, 2001; Thornton et al., 2016; Van Der Wiel et al., 2019). For instance, Texas was hit in February 2021 by a severe cold snap that led to cascading failures in the energy system, resulting in a power outage and leaving millions of Texans without electricity (Busby et al., 2021). While those temperatures were extreme, they were not unprecedented. However, population growth and ongoing electrification made the system more vulnerable to such extreme events (Doss-Gollin et al., 2021).





In France, the historical cold spell of January 1985 led to peaks in electricity demand and power outages (Caud and Vautard, 2018; Le Monde, 1985; RTE, 2021). More recently, the cold spell of February 2012 caused a record consumption of electricity and put the energy network at risk (Le Monde, 2012a, b). In 2023, the electricity transmission system operator Réseau de Transport d'Électricité (RTE) warned ahead of the winter season of several reasons for tension in the electricity supply — lack of gas supply due to the economic and geopolitical context combined with the unavailability of parts of the nuclear power plants

for technical reasons — potentially leading to outages in the event of high demand caused by a cold spell (RTE, 2023b). The resulting reduction in electricity consumption and the overall mild winter allowed avoiding power outages in France. However, RTE stated that up to 12 red Écowatt signals (power cut inevitable without a reduction of consumption) could have been raised otherwise (RTE, 2023a). Rouges et al. (2024) recently analyzed the impact of atmospheric patterns on energy shortfalls in Europe, due to cold temperatures and the lack of wind. Therefore, even in a warmer world, cold events remain a main concern

for the energy systems, especially in case of increased vulnerability or in facing an energy crisis (RTE, 2021).

    As the lower atmosphere warms due to climate change, it is expected that cold extremes will decrease (Seneviratne et al., 2021). Warming trends have already been detected in various indicators, such as cool nights (TN10p) and the coldest night of the year (TNn), as reported in observations and reanalysis studies (Donat et al., 2016; Morak et al., 2013). Statistical analyses also show a faster decrease in all-time daily low records than compared to a stationary climate (Finkel and Katz, 2018).

Attribution studies on specific events, such as the winter of 2010 in Europe, have shown that these recent extreme events would have been even more extreme without climate change (Cattiaux et al., 2010; Christiansen et al., 2018). Similar analyses performed on climate model simulations to investigate the evolution of cold events in the future also indicate a continuation of the frequency decrease in cold extremes from several key indicators (Coppola et al., 2021; Gross et al., 2020; Kim et al., 2020; Thorarinsdottir et al., 2020; Wehner et al., 2020).

From a physical point of view, winter cold spells in Western Europe are typically caused by a blocking event over Greenland, the North Atlantic, or Scandinavia, which disrupts the prevailing westerly flow into Europe and allows the advection of cold air from the Arctic and Russia (Bieli et al., 2015; Buehler et al., 2011; Brunner et al., 2018; Pfahl and Wernli, 2012; Pfahl, 2014; Sillmann et al., 2011; Sousa et al., 2018). This blocking is often associated with a large cyclone over the Adriatic and Ionian seas. Although the existence of the blocking is crucial, its localization can vary. European cold winter extremes are

often large-scale events affecting different regions simultaneously, which explains why the North Atlantic Oscillation (NAO) is a good indicator of cold extremes in Western Europe. A persistent negative phase of the NAO (NAO−) is often linked to long-lasting atmospheric blocking in the North Atlantic, with a cold anomaly located downstream or south of the blocking (Cattiaux et al., 2010; Greatbatch, 2000; Kautz et al., 2022; Pfahl, 2014; Seager et al., 2010; Sillmann et al., 2011; Thompson and Wallace, 2001; Wang et al., 2010).

Nevertheless, there are uncertainties surrounding the impact of Arctic amplification and the Atlantic Meridional Overturning Circulation (AMOC) on cold extremes. For instance, these phenomena could increase the meanders of the jet stream, leading to more frequent and intense cold spells (Blackport and Screen, 2020; Geen et al., 2023).

    The inherently low occurrence of very extreme events makes them difficult to study because of the resulting lack of samples. For instance, ≈3000 years of data are needed to have a 95% probability of having at least one occurrence of a millennial event:



$P_n(X \geq 1) = 0.95$ for $n = 2994$, where $X$ is the number of occurrences of an event with a yearly probability of $p = 1/1000$ and $n$ the number of years during which that event could occur. So if we consider a 50-year climate period, even a large simulation ensemble of up to 50 members is not sufficient to ensure the occurrence of at least one millennial extreme event.

To address this, various methods rooted in statistical physics have been developed to simulate realistic extreme atmospheric variables. Rare events algorithms based on importance sampling (e.g., Ragone et al., 2018; Ragone and Bouchet, 2021) have
been developed to specifically simulate extreme heatwaves from climate models by selecting and cloning trajectories that are most likely to lead to extremes. Gessner et al. (2021) proposed a similar approach (so-called "ensemble boosting"), selectively targeting climate model trajectories yielding higher temperatures and running new ensembles of perturbed reinitialized clones at the starting points of these trajectories. Sippel et al. (2023) adapted this method to extreme cold winters over a specific region (Germany). Finkel and O'Gorman (2024) demonstrated that ensemble boosting is quite optimal to simulate short-lived (up to
2 weeks) extreme events.

Another approach employs Stochastic Weather Generators (SWGs), which are Markov processes generating large ensembles of atmospheric trajectories with realistic statistics at minimal computational expense (Ailliot et al., 2015). Yiou and Jézéquel (2020) integrated a SWG based on circulation analogues (Yiou, 2014) with importance sampling to specifically simulate extreme summer heatwaves from circulation analogues, enabling physically consistent trajectories of extreme events at very
low computational cost.

Those algorithms were mainly developed and used to study summer heatwaves, but the ensemble boosting and the SWG have both been adapted to extreme cold winter events by Cadiou and Yiou (2024) and Sippel et al. (2023). Nevertheless, those results were only applied to present-day events and limited to ERA5 reanalysis and the CESM2 climate model. Here, we apply the SWG to various CMIP6 model outputs to have a broader view of low-probability, high-impact extreme cold spells in the
future. We then analyze the resulting simulations to evaluate potential changes in the intensity and dynamics of extreme cold spells in the future. Short events (1–2 weeks) are more relevant in terms of associated impacts on the energy sector (Añel et al., 2017; RTE, 2021, 2023b), therefore we focus in this study on persisting cold spells of 15 days.

Rare event algorithms primarily aim to simulate extreme events. Their nudging or score function typically maximizes the variable of interest, such as temperature, precipitation, or wind speed. However, we argue that using them with another nudging
variable or score function — that does not directly involve the variable of interest, like temperature — allows for identifying the drivers of the extremes of the variable of interest. For instance, Noyelle (2024, Chap. 6) applied a rare event algorithm using as score functions soil moisture or geopotential height at 500 hPa at a grid point, to investigate their effect on high temperatures. Similarly, in this study, we further analyze the link between North Atlantic atmospheric dynamics and extreme cold spells in France by running SWG simulations with empirical importance sampling on the dynamics instead of temperature, in order to
isolate the effect of dynamics on the extremeness of the event.

This paper is organized as follows: Section 2 presents the data used in the study. Section 3 presents the statistical model used and the various experiments conducted with it. Results are presented in Section 4. First, we evaluate how the intensity and dynamics of extreme cold spells evolve with climate change in CMIP6 models. Then, we investigate more precisely the role of dynamics in extreme cold spells in both the present and future. Finally, discussions and conclusions appear in Section 5.





## 2 Data


In this paper, we investigate the evolution of cold spells in France in the ERA5 reanalysis dataset (Hersbach et al., 2020) and in an ensemble of CMIP6 model simulations (Eyring et al., 2016). ERA5 was chosen for its extensive time coverage (from 1950 to the present day) and its high horizontal resolution of 0.25°. CMIP6 was chosen to allow the intercomparison of several Global Climate Models (GCMs) for future projections.

The domain of interest for temperature is metropolitan France. We consider daily average temperature (TG). The daily temperature values of ERA5 and CMIP6 are interpolated on the Système d'Analyse Fournissant des Renseignements Adaptés à la Nivologie (SAFRAN Vidal et al., 2010) reanalysis grid, with a horizontal resolution of 8 km. Then a mask of metropolitan France is applied to have a more accurate weighting of continental surfaces. We then compute the daily spatial average of temperature over metropolitan France.

CMIP6 models all yield temperature biases (cf. Fig. 1). There are many sophisticated ways of correcting model biases (François et al., 2020), but those methods are generally not appropriate for extremes. Therefore, we simply apply a first-order bias correction by removing the difference in medians of DJF temperature means from 1950 to 2000 (as displayed in Fig. 1) between each model and ERA5. This avoids the generation of artefacts on extremes due to bias correction methodologies.

To characterize the atmospheric circulation, we use the mean daily geopotential height at 500 hPa (Z500). The preference for 110   Z500 over sea-level pressure (SLP) is motivated by its lower dependence on disturbances originating from surface roughness, as well as its widespread utilization in the study of weather regimes, as substantiated by several studies (Corti et al., 1999; Dawson et al., 2012; Jézéquel et al., 2018; Yiou and Nogaj, 2004). (Jézéquel et al., 2018) have highlighted that Z500 is more suited for simulating European temperature anomalies, even though their investigation was centred on warm summer temperatures.

We selected eleven (11) models out of 34 CMIP6 models. The selection criterion is the availability of daily temperature and 115   Z500 data on the IPSL meso-center. For simplicity, we considered only one model run when ensembles were available. The resulting list of models is specified in Table 1. The horizontal resolutions range from 100 km to 500 km.

To provide a first-order evaluation of the accuracy of these models for winter temperatures over France, we compute the December to February (DJF) temperature averages (over metropolitan France) for each winter from 1950-1951 to 1999-2000 and compare the distributions with ERA5. The bias is then calculated as the difference between the median of the model and 120   ERA5. Results are presented in Figure 1 and Table 1. The mean DJF bias ranges from −1.43°C to +3.05°C, with 7 out of 11 models displaying a positive bias and 4 exhibiting a negative bias with respect to ERA5. CNRM-ESM2-1, KACE-1-0-G, FGOALS-g3, and IPSL-CM6A-LR exhibit the lowest bias (in absolute value). We performed a Kolmogorov-Smirnov test of the similarity between the winter temperature distributions of ERA5 and each CMIP6 model simulation (the null hypothesis is that the distributions are the same). This null hypothesis could not be rejected for those four models with the lowest biases 125   (($p > 0.05$). The computed bias was removed from daily temperature data for each model. This performs a first-order bias correction as mentioned at the beginning of this section. This bias estimation offers an initial approximation of the model accuracy for winter temperatures in France. We use it as a correction to allow a better comparison to ERA5 data, but it does not necessarily indicate that a model with low bias is proficient for accurately simulating extreme cold spells in France.



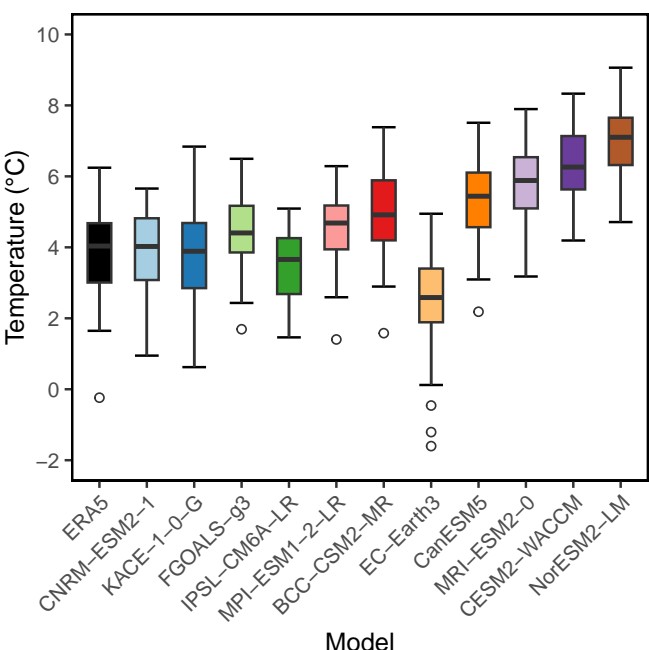

**Figure 1.** Empirical probability distributions of DJF mean temperatures (without mean correction) over France from 1950 to 2000 for ERA5 (black) and the 11 CMIP6 models (colours) in Table 1.

| Model name | Institute ID | Approx. Horiz. Res. | Run label | Reference | Bias (°C) | TG15d min. |
|---|---|---|---|---|---|---|
| CNRM-ESM2-1 | CNRM | 250 km | r1i1p1f2 | Séférian et al. (2019) | −0.13 | 1961-01-16 |
| KACE-1-0-G | NIMS-KMA | 250 km | r1i1p1f1 | Lee et al. (2020) | −0.17 | 1978-02-06 |
| FGOALS-g3 | CAS | 250 km | r1i1p1f1 | Li et al. (2020) | 0.31 | 1983-01-28 |
| IPSL-CM6A-LR | IPSL | 250 km | r1i1p1f1 | Boucher et al. (2020) | −0.41 | 1952-01-03 |
| MPI-ESM1-2-LR | MPI-M | 250 km | r1i1p1f1 | Mauritsen et al. (2019) | 0.61 | 1954-12-30 |
| BCC-CSM2-MR | BCC | 100 km | r1i1p1f1 | Wu et al. (2021) | 0.92 | 1962-12-27 |
| EC-Earth3 | EC-Earth-Consortium | 100 km | r1i1p1f1 | Döscher et al. (2022) | −1.43 | 1973-12-28 |
| CanESM5 | CCCma | 500 km | r1i1p1f1 | Swart et al. (2019) | 1.46 | 1971-01-12 |
| MRI-ESM2-0 | MRI | 100 km | r1i1p1f1 | Yukimoto et al. (2019) | 1.88 | 1982-01-22 |
| CESM2-WACCM | NCAR | 100 km | r1i1p1f1 | Danabasoglu et al. (2020) | 2.21 | 1977-01-05 |
| NorESM2-LM | NCC | 250 km | r1i1p1f1 | Seland et al. (2020) | 3.05 | 1981-02-09 |

**Table 1.** List of selected CMIP6 model runs. The selection criterion is the availability of daily data for Z500 and temperature on the IPSL computing server. The bias is computed as the difference in medians of DJF temperature means over France from 1950 to 2000 between each model and ERA5. The last column indicates the date of occurrence of the coldest TG15d event in the historical simulation (1950-2014).



To investigate the projected climate until 2100, we selected four Shared Socioeconomic Pathways (SSPs: Riahi et al. (2017))

to cover the broad range of future plausible socioeconomic and climatic scenarios: SSP1-2.6, SSP2-4.5, SSP3-7.0, SSP5-8.5. For each model, we computed the global mean surface temperature (GMST), and its yearly increase from the 1950-2000 average.

To compute atmospheric circulation analogues, we use Z500 anomalies over the North-Atlantic spatial domain (20°W – 30°E; 30°N – 70°N) as outlined by Cadiou and Yiou (2024). Anomalies of Z500 are computed on a daily basis, subtracting

the seasonal cycle as a 31-day moving average computed on the 1981-2010 reference period. Z500 increases as a result of the warming of the atmosphere due to the expansion of the lower atmosphere. Therefore a linear trend is removed in Z500 before computing the analogues for each model and scenario.

## 3    Methods

### 3.1    Analogues-SWG

We first establish a database of circulation analogues, following the methodology outlined by Yiou and Jézéquel (2020). For any specific day $t$, we calculate the (spatial) Euclidean distance between the Z500 fields of day $t$ and all other days $t'$, ensuring these days do not fall within the same year or season (straddling two consecutive years) and maintain a calendar distance from $t$ that is less than 30 days (i.e., $|t - t'| \leq 30$ days). The $K$ best analogue days for $t$ are identified as the $K$ days with the minimum distance from $t$. In line with recommendations from prior research (Platzer et al., 2021), we select $K = 20$ best analogues.

To study the evolution of cold spell atmospheric dynamics from 1950 to 2100, we computed circulation analogues in three non-overlapping periods of 50 years:

1. 1950-1999: past period over which we can compare CMIP6 and ERA5 data.

2. 2000-2049: present-near future period in CMIP6. The rationale for considering this period is that SSP scenarios barely differ before 2050.

3. 2050-2099: future period in CMIP6.

For CMIP6 models, the historical period (1950–2015) is concatenated to the run of each SSP (2015–2100). This results in a 150-year files for each model and SSP, which is then divided into three non-overlapping climate periods of 50-year. Therefore the first analogue period (1950-1999) is common to the four SSPs, while separate sets of analogues are computed over 2000-2049 and 2050-2099 for each SSP. The period 1950-1999 is used for comparison with ERA5.

The stochastic weather generator (SWG) developed by Yiou (2014) uses circulation analogues to generate alternative trajectories of temperature or precipitation values by reshuffling daily atmospheric fields. This algorithm was modified by Yiou and Jézéquel (2020) to simulate extreme heat waves using importance sampling and by Cadiou and Yiou (2024) to simulate extreme cold winter events from the ERA5 reanalysis. As we want to simulate extreme cold spells, we hereafter use the version of the SWG developed by Cadiou and Yiou (2024).





The SWG starts at a given initial condition and proceeds to the next time step using analogues of circulation and information on their "next day". The simulation is a constrained reshuffling of days from the input dataset. At each time step, the selection of the analogue day is subject to constraints and weights that are controlled by model parameters explained below, so that the sampling is not necessarily uniform among the $K$ best analogues.

To follow the seasonal cycle, $K$ weights $\omega_{cal}^{(k)}$ are used, depending on a parameter $\alpha_{cal}$ that favours analogue days closest to
the calendar date of time step $t$:

$$\omega_{cal}^{(k)} = A_{cal}e^{-\alpha_{cal}d_k} \tag{1}$$

To simulate the most extreme events, importance sampling weights $\omega_T^{(k)}$ are introduced, with a control parameter $\alpha_T \geq 0$. The higher $\alpha_T$, the more the SWG favours analogue days with extreme temperatures. The $K$ analogues of $t$ are sorted in ascending order of temperature with ranks $R_k \in \{1, \ldots, K\}$:

$$\omega_T^{(k)} = A_T e^{-\alpha_T R_k} \tag{2}$$

Note that the $K$ values of $\omega_T^{(k)}$ are the same from one time step to the next because they do not depend on the temperature
value, but on the rank of the circulation analogues temperature.

The SWG with importance sampling is achieved by combining the calendar and importance sampling weights. The probability of sampling the $k^{th}$ analogue day of day $t$ is given by:

$$\omega_k = Ae^{-\alpha_{cal}d_k}e^{-\alpha_T R_k} \tag{3}$$

where $A$ is a normalizing constant ensuring that the sum of all probabilities $\omega_k$ equals 1.

Finally, we use the SWG configuration established by Yiou and Jézéquel (2020): at every time step, the weights $\omega_k$ of
analogue days that occur within the observed event are assigned a value of zero. This ensures that the simulations are driven exclusively by the initial conditions and the climate state, without relying on any specific information about the observed event (except for the first day). In essence, we are evaluating whether the occurrence of a record-shattering event can be inferred from information related to less extreme events.

### 3.2    Protocol of SWG simulations

To evaluate the impact of climate change on extreme 15-day cold spells, we make SWG simulations of extreme cold events from the three different climate periods and the four SSPs, i.e. using the analogue sets computed for each. For each model, we identify the coldest TG15d in the historical period (1950-1999). Those dates are indicated in Table 1. Then we initialize the SWG at the starting date of that event and run 1000 simulations per analogue period (1950-1999, 2000-2049 and 2050-2099) and scenario (SSP1-2.6, SSP2-4.5, SSP3-7.0, SSP5-8.5). Thus, $1000 \times 3 \times 4$ simulations are made for each of the 11 climate
models. Simulations made in 1950-1999 should be very similar across SSPs as the analogue set used is the same. This SWG simulation protocol is an innovation compared to the simulations of Cadiou and Yiou (2024), as non-overlapping analogue periods are considered, as well as a multi-model ensemble, rather than just a reanalysis. For comparison over the historical



period, we perform 1000 simulations starting on the 3rd of January 1985 (beginning of the coldest TG15d in ERA5) using ERA5 analogues from 1950-1999.

This SWG approach of running alternative trajectories of pre-existing extreme events is similar to the "ensemble boosting" method of Gessner et al. (2021). The ensemble boosting approach starts by identifying the most extreme events in a long control simulation of a climate model. Then the model is initiated a few days before the apex of the extreme, and ensembles are run from small perturbations of the initial condition (Gessner et al., 2021). The ensemble boosting procedure keeps the most extreme simulations. It has been argued that such an approach is optimal to simulate short-lived extremes (Finkel and

O'Gorman, 2024). The SWG approach was previously applied to the study of potential future heatwaves by Yiou et al. (2023), who ran ensembles of simulations starting before an identified heatwave in each model for different SSPs. The main difference with the study of Yiou et al. (2023) is that major cold spells occurred in the 20th century, while the most intense heatwaves are bound to occur toward the end of the 21st century.

The same approach was also used by Cadiou and Yiou (2024), using the single observed event of winter 1963 as a reference.

That study focused on whole winter events over the available period of ERA5 data (1950-2021). Here, we use the SWG over a wide range of climate model data and SSPs to gain a broader view of the evolution of 15-day cold spells in the future, both in terms of intensity and dynamics.

### 3.3   Atmospheric circulation index for cold spells

Cold spells in France are generally associated to an easterly or northeasterly flow of cold air, which can be either dry or humid

(e.g. Yiou and Nogaj, 2004). These flows are typically caused by an anticyclone around Iceland or Scandinavia, which leads to a weakening of the westerlies and allows the outbreaks of cold air. These weather patterns can persist for up to a week or ten days, with temperatures dropping significantly, especially during nighttime clearings on snow-covered grounds in humid episodes. These mechanisms are strengthened by an anticyclone over southern Europe.

Z500 mean anomalies maps during the coldest 15-day cold spells in ERA5 show a dipole between Iceland and southwestern

Europe. Therefore, to investigate the dynamics of cold spells in France, we compute a Western Europe Cold Circulation index (WCC), which characterizes this atmospheric pattern, by subtracting the mean of Z500 between (1°W – 9°E; 40°N – 47°N) and (24°W – 13°W; 62°N – 66°N). Those areas were chosen as the maximum and minimum of the dipole structure identified from the Z500 composite map of the 20 coldest cold spells in France from ERA5. As shown in Fig. 6, the WCC is correlated to daily temperature over France during winter months (Pearson correlation coefficient $r = 0.65$ and $p < 10^{-15}$). As this WCC

index is tailored for a specific type of event (15-day cold spells) over a specific region (metropolitan France), it performs better than a classic daily North Atlantic Oscillation (NAO) index, defined as the normalized SLP difference between the Azores and Iceland ($r = 0.36$ and $p < 10^{-15}$).

To facilitate the comparison between models, the WCC index was normalized to $(-1, 1)$. The normalization was done by subtracting the mean value and dividing by the range (max minus min) of the data:

$$x_{normalized} = \frac{x - x_{mean}}{x_{max} - x_{min}}, \qquad (4)$$



where $x$ is the Z500 difference. Therefore an index of $-1$ corresponds to the most extreme dipole observed in ERA5, characterized by a strong high-pressure system over Iceland and a strong low-pressure system over southwestern Europe. An index of 1 corresponds to the opposite configuration, with a low-pressure anomaly over Iceland and a high-pressure anomaly over southwestern Europe. This normalization allows for the identification of atmospheric configurations that tend towards the identified dipole, with more negative values indicating a stronger and more intense pattern.

### 3.4 SWG with importance sampling on circulation


After analysing the atmospheric dynamics of extreme cold spells and its evolution between 1950 and 2100, we want to evaluate the role of the atmospheric circulation in triggering cold spells over France. To this end, we run simulations of the SWG, applying the importance sampling to WCC instead of temperature. We call these new simulations WCC-SWG, as opposed to T-SWG where the importance sampling is on temperature. In this way, no direct incentive is given to temperature in the

SWG simulations. The SWG is only parameterized to favour trajectories with a low WCC, i.e., with a Z500 configuration tending towards a dipole featuring a high over Iceland and a low over Western Europe. This allows us to identify the effect of this atmospheric pattern on temperatures in France. This aims to show to what extent the identified atmospheric circulation is *sufficient* to trigger extreme cold spells over France. This analysis is also an innovation from the paper of Cadiou and Yiou (2024).

Results are compared with SWG simulations putting the importance sampling over temperature (as in previous simulations), a simple NAO index (NAOi: computed between the two gridpoints closest to the stations located in Iceland and the Azores as in Rogers (1984)), and no importance sampling ($\alpha_T = 0$). They will be referred as T-SWG, NAO-SWG and control-SWG respectively.

For each model, we also fit an autoregressive model of the first order ($AR(1)$) on WCC anomalies over winter months,

which acts as a control:

$$X_t = c + \varphi X_{t-1} + \epsilon_t \tag{5}$$

where $c$ is a constant and $\epsilon_t$ is a centered Gaussian white noise with standard deviation $\sigma$ ($\epsilon_i \sim N(0, \sigma^2)$). The parameters $\phi$ and $\sigma$ are estimated in a standard way by assuming that $X_t$ and the WCC variations have the same variance and auto-covariance at lag 1 (Storch and Zwiers, 2002). The numerical values of estimated parameters are computed for each model and scenario. For instance for KACE-1-0-G and SSP1-2.6 $\hat{\varphi} = 0.859$ $\hat{\sigma} = 0.151$ and $c = -0.316$. The AR(1) has the same autocovariance as the

winter WCC of each model but is not a measure of atmospheric circulation, so it should not have any effect on temperatures. We then simulate SWG trajectories that minimize the AR(1) values (AR-SWG simulations).

This procedure defines an alternative way to causal networks (Kretschmer et al., 2021) to explore a causal relation between an atmospheric feature and extreme cold spells in France. The principle of this simple approach is to nudge towards extremes several "candidates" for causality (here several atmospheric indices) and determine the effect on the extremes of a climate

variable (here temperature). This reflects the spirit of the "do" action outlined by Hannart et al. (2016).





## 4 Results

### 4.1 Intensity of winter cold spells in CMIP6

#### 4.1.1 Extreme TG15d in model output

First, we examine the five coldest TG15d events in ERA5 and each of the 11 CMIP6 models across the historical period and
the four SSPs, all merged together (Fig. 2a). We find that only one model (EC-Earth3) is able to simulate extreme TG15d cold
spells as cold as those observed in ERA5. The other models are between 0 and 3°C above such low temperatures. EC-Earth3
simulates an event with a temperature of $-8,1°C$ on 28/12/1973, while the coldest event recorded in ERA5 is $-7.5°C$ on
01/10/1985. Despite the upward temporal trend in temperatures in all models, some models still simulate extreme cold TG15d
events in high emissions scenarios or late in the century. For example, MPI-ESM1-2-LR simulates an event with a temperature
of $-5.8°C$ on 20/10/2048 under SSP5-8.5, while IPSL-CM6A-LR simulates an event with a temperature of $-4.9°C$ on the
18/01/2060 under SSP1-2.6. Although those events occur in a warmer climate, they are still comparable to the coldest events
recorded in ERA5 at the end of the 20th century, such as $-5.0°C$ on the 15/01/1987 or $-4.8°C$ on the 29/01/1963. However,
they are more than $2°C$ warmer than the coldest event recorded in ERA5 ($-7.5°C$). No model can produce colder events after
the historical period (beyond 2015).
We plotted the five coldest cold spells by level of warming compared to 1950-1999 (Fig. 2b). We show that half of the coldest
cold spells occur for levels of warming under $0.2°C$ compared to 1950-2000. A few high-intensity events are still found for
higher levels of warming (TG15d of $-4.9°C$ in IPSL-CM6A-LR for a warming of $1.7°C$, and a cold spell reaching $-5.8°C$ for
a warming of $1.6°C$ compared to 1950-2000 in MPI-ESM1-2-LR). However, for all models, none of the 5 most intense cold
spells happen for a level of warming higher than $1.8°C$ compared to 1950-2000.

#### 270 4.1.2 Extreme TG15d in CMIP6 with a SWG

To explore the evolution in terms of intensity and dynamics of extreme cold spells in the future, we run SWG simulations from
CMIP6 data from 1950 to 2100, as described in Section 3.2. For each model, we run simulations starting at the beginning of
the coldest event found in the model run over the historical period of 1950-1999 (see Table 1). We produce 1000 simulations
for each of the 50-year periods and each of the four SSPs. The discrepancies between models observed in the TG15d cold
spells simulated by the SWG are similar to those found in the cold events detected in the "raw" model runs (Fig. 3). In the
historical period (1950-1999), the median temperature of the SWG simulations varies significantly, with the coldest model
(EC-Earth3) reaching a temperature of $-3.8°C$ and the warmest model (FGOALS-g3) reaching a temperature of $0.4°C$. In
comparison, the median temperature of the ERA5 simulations for the same period is $-2.27°C$. These results indicate that there
are substantial differences between the models, even in the historical period, and that most of the models are warmer than the
ERA5 simulations even after correcting by the historical DJF mean. Only the SWG simulations from KACE-1-0-G manage to
display a distribution that is comparable to ERA5 SWG simulations according to a Kolmogorov-Smirnov test. Thus, the results
in the following sections will focus on this model.





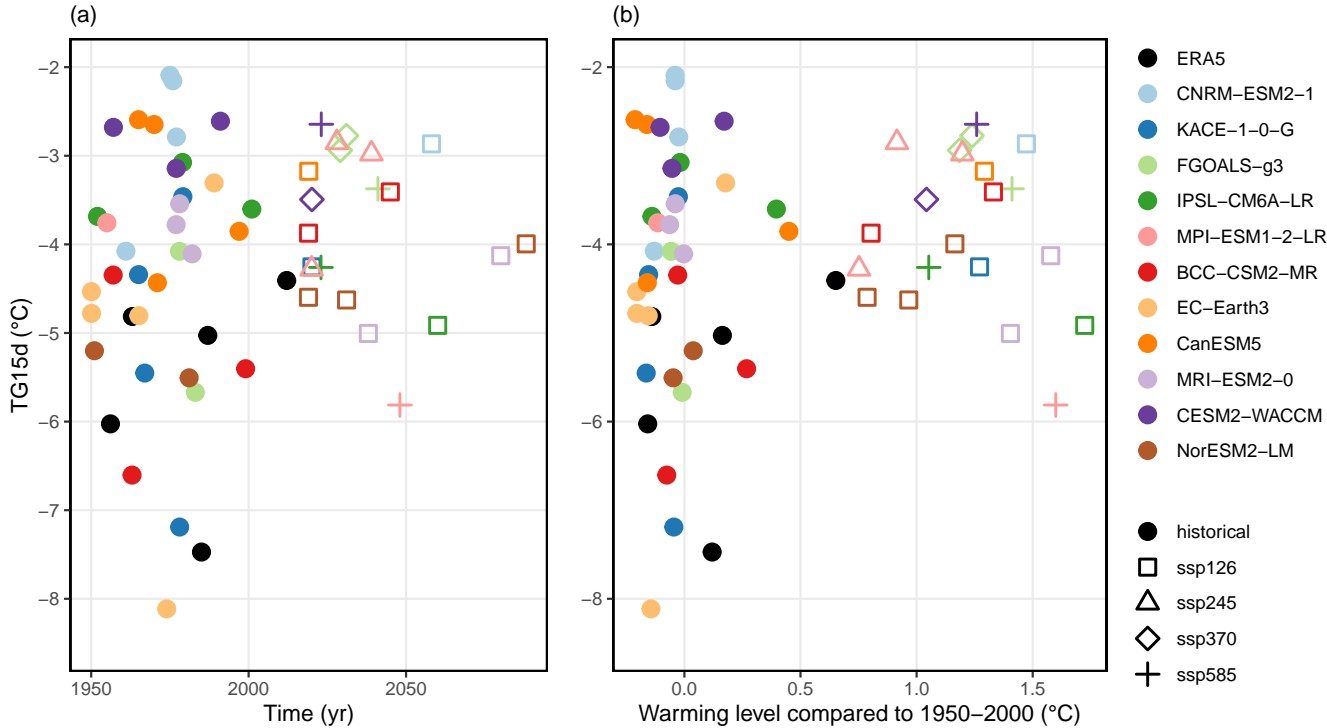

**Figure 2.** Five coldest 15-day 2-m temperature mean (TG15d) in ERA5 from 1950 to 2023 (black circles) and in 11 CMIP6 models (colours) from 1950 to 2100 across the historical period and the four emissions scenarios (shapes) (a) by date and (b) by level of warming compared to 1950-2000. Temperatures are adjusted by the median DJF temperature bias.

Out of the eleven models examined, only three (EC-Earth3, KACE-1-0-G, and BCC-CSM2-MR) allowed SWG simulations of TG15d events colder than the coldest ERA5 event in the historical period. This implies that the remaining models may be

underestimating the potential intensity of future cold waves. However, we note that the models which produced the coldest events in the historical period also exhibited the fastest warming rates. Consequently, the disparity in the extreme cold events simulated by the different models at the end of the 21st century is not as wide. For instance, under SSP5.8-5, the standard deviation of the simulations for the period 1950-1999 is $1.83°C$, compared to $1.47°C$ for the period 2050-2099.

In the present-day period (2000-2049), the impact of climate change on cold spells is limited across all scenarios, as more

than half (between 6 or 7 out of 11 depending on the scenario) of the models still simulate extreme cold events that are colder than the ERA5 median for the period 1950-1999. However, as the level of warming increases towards the end of the 21st century in high-emissions scenarios, the likelihood of very extreme cold events, comparable to the cold events of 1963 or 1985 in the 21st century, becomes negligible in all scenarios. The coldest event simulated by the SWG in the period 2050-2099 under the SSP5-8.5 scenario, obtained from CESM2-WACCM analogues, reaches a temperature of $-1.69°C$ over France. This

temperature is warmer than the median of ERA5 SWG historical simulations for the same region.



**Figure 3.** SWG simulations of 15-day cold spells for ERA5 (black, only 1950-1999 analogue period) and CMIP6 models (colours), for four SSPs ((a) SSP1-2.6, (b) SSP2-4.5, (c) SSP 3-7.0, (d) SSP5-8.5) and three analogues periods (1950-1999, 2000-2049, 2050-2099). Each boxplot displays 1000 simulations. Temperatures are adjusted by the median DJF temperature bias. In the box plots, the boxes represent the median ($q50$), with the lower and upper hinges denoting the first ($q25$) and third ($q75$) quartiles, respectively. The upper whiskers are defined as $\min[\max(T), q75 + 1.5 \times (q75 - q25)]$, while the lower whiskers are formulated symmetrically. The individual points in the plot correspond to the outlying values that exceed the upper whisker or fall below the lower whisker.



## 4.2 Dynamics of winter cold spells

### 4.2.1 Atmospheric dynamics of very intense cold spells in CMIP6

To investigate whether models reproduce the same atmospheric dynamics leading to cold spells as in ERA5, we computed the WCC index for each model and scenario. In all 11 CMIP6 models, the daily winter temperature over France is significantly correlated with the WCC (p-value $< 10^{-15}$), although the correlation is not as strong as in ERA5. The correlation ranges from 0.45 in CanESM5 (which is the model with the coarsest resolution) to 0.56 in CESM-WACCM and MRI-ESM2-0. If a 1-day shift is considered between the WCC and temperature, the correlation is higher, ranging from 0.49 for CanESM5 to 0.61 for CESM-WACCM. This suggests that the WCC measures how the atmospheric circulation causes the advection of cold air into France, hence triggering cold spells. Correlation does not imply causality and essentially measures the covariations of "small" fluctuations of temperature and WCC. Here, we are interested in the correspondence between temperature and WCC in the extremes, which might not be connected to the mean correlation.

We first determine the distribution of WCC probability distribution for the events previously simulated with the SWG. This helps us assess whether a low value of WCC is a *necessary* condition for cold spells (i.e. the behaviour of WCC when TG15d is cold). The results are shown in Figure 6. For most models, the identified cold spell episodes lead to negative values of WCC, indicating that they are associated with the identified dipole of Z500. However, some models (CanESM5, FGOALS-g3, NorESM2-LM) have mostly positive WCC values, suggesting that they have difficulty reproducing the same patterns that lead to very extreme cold spells as observed in ERA5. For instance, CanESM5 is also the model that had the weakest correlation between the WCC and winter daily temperature over France, which can be explained by its coarse resolution ($\sim 500$ km). Those models still produce a similar dipole pattern in the Z500 field during the events they simulate. However, the high-pressure centre in the dipole is in some models shifted to the south of Iceland (IPSL-CM6A-LR) or towards Scandinavia (NorESM2-LM or EC-Earth3 for instance), and the low-pressure centre is shifted to the west (FGOALS-g3), compared to the dipole pattern observed in ERA5 (See Supplementary Materials).

The maps (Fig. 4) and the WCC values (Fig. 6) show that there is no significant change in the dynamics of cold spells with high levels of warming. In Fig. 6 more than half of models have a negative median WCC in SWG simulations. Models that have the most positive values are CanESM5, NorESM2-LM and FGOALS-g3, which are the models that struggled the most in producing extreme cold events comparable to ERA5 with the SWG (see Fig. 3). We also note that CanESM5 is the model with the coarsest resolution and NorESM2-LM is the model that had the largest DJF temperature bias (Fig. 1). KACE-1-0-G, the model with extreme TG15d temperature the closest to ERA5 in the historical period, has the majority of its SWG simulations with negative WCC whatever the period or SPP. The values of the WCC are fairly consistent across periods and scenarios within each single model. This suggests that the mechanisms driving these events are not significantly affected by climate change in the CMIP6 models that we consider. Even if global warming affects the intensity of this type of events, it does not cause major changes in their dynamics in those models.

To ensure that the composite maps are representative of the dynamics of individual simulations, we computed the standardized standard deviation of Z500 from the $10\%$ coldest SWG simulations, for each grid point. The standard deviation was





calculated from the 100 coldest simulations (out of 1000 SWG simulations) and normalized by the climatological standard deviation of Z500 for each climate model. This adjustment for each gridpoint was necessary because Z500 exhibits greater variability in the high latitudes than in the low latitudes. The results for KACE-1-0-G are presented in Fig. 5. The simulations demonstrate a high degree of similarity of Z500 maps over Europe and the North-East Atlantic between SWG simulations. This outlines the relevant atmospheric features to trigger winter cold spells in France. Those areas are very consistent across

periods, scenarios and models (see Supplementary materials for additional figures). Some models do have a larger standard deviation between simulations (FGOALS-g3 or CanESM5) but most of the domain still has a standardized standard deviation under 1, which means that simulations are more alike than random days.

### 4.2.2  Role of atmospheric conditions to generate cold spells

In this section, we evaluate how an atmospheric circulation pattern leads to a cold event. We have investigated in the previous

section the mean atmospheric patterns that prevail during cold spells. This corresponds to assessing the necessary atmospheric patterns for a cold spell. Conversely, we now evaluate how those atmospheric patterns lead to extreme cold spells, which corresponds to a sufficient condition. Such a sufficient condition can be anticipated by the 1-day lag between the WCC index and temperature. Here, we verify that this relation holds for the most extreme events.

We test this hypothesis by running SWG simulations with importance sampling weights on the WCC index (WCC-SWG),

rather than temperature (T-SWG). Hence, the SWG simulations exacerbate the effects of this bipolar atmospheric pattern. No nudging is added on temperature, so that the sole driver of the temperature simulated by the SWG is the atmospheric circulation. For comparison purposes, SWG simulations are also performed with importance sampling weights on temperature (as in Section 4.1.2), an NAO index (NAO-SWG), an AR(1) process (defined in 3.4, AR-SWG), and with no importance sampling (control-SWG). Results are shown for the KACE-1-0-G model and SSP5-8.5 in Fig. 7. The results for other models and SSPs

are shown in the Supplementary Materials. Very consistently across models, scenarios and periods, WCC-SWG simulations yield temperatures that are as cold as T-SWG simulations, even for models that did not accurately reproduce the WCC distribution. In comparison, AR-SWG and control-SWG simulations produce events with milder temperatures, corresponding to an average TG15d event in winter for each period and scenario. NAO-SWG simulations are overall colder than AR-SWG and control-SWG but do not reach cold temperatures of T-SWG simulations. This demonstrates that atmospheric circulation is the

main driver of cold extremes in France and that an index tailored for this region and event type performs well in capturing the circulation associated with these extreme events.

Therefore a dipole featuring a high over Iceland and a low over Western Europe is a sufficient condition to trigger extreme 15-day cold spells over Western Europe in the selected CMIP6 models, even in a climate with a higher level of warming.

To assess the evolution of the atmospheric pattern, we first determine the $5^{th}$ percentile of the winter daily WCC for the

historical period of each model. For each year between 1950 and 2100, we then compute the number of winter days with WCC falling under that threshold. We compute a yearly linear regression of this number of days. The trend in the number of days that fall below this percentile is computed for each model and scenario. The significance of the trend is assessed by a Mann-Kendall test (Storch and Zwiers, 2002). As depicted in Fig. 8, about half of the models do *not* exhibit a significant trend.



**Figure 4.** Absolute values (contours, in m) and anomalies (shaded areas, in m) with respect to 1950-1999 of 500-hPa geopotential height (Z500) for the 10% coldest SWG simulations (i.e. 100 trajectories) for each period (columns) and SSP (rows) in KACE-1-0-G. Each maps displays the adjusted composite temperature on the top left and the composite normalized WCC on the top right.

Between models that *do* yield a trend, there is a substantial disparity in the direction of potential trends. However, the trends are largely consistent across scenarios for a single model, indicating that the dynamic evolution is coherent within individual models.



**Figure 5.** Standardized standard deviation (shaded areas, $\sigma$) and anomalies with respect to 1950-1999 standard deviation of 500-hPa geopotential height (Z500) for the 10% coldest SWG simulations (i.e. 100 trajectories) for each period (columns) and SSP (rows) in KACE-1-0-G.

We find that 14 runs (from 6 different models, as we consider 4 SSPs per model) yield a significant negative trend, while only 6 runs (from 2 models) exhibit a positive trend. Therefore, among the models selected for this study, the majority of models exhibit a negative trend of low WCC, although those trends are not necessarily significant. However, the GCM whose extreme cold spells in the historical period are the closest to ERA5 (KACE-1-0-G) does not yield any significant trend in any of the




**Figure 6.** Normalized WCC composites of the SWG output simulations for each CMIP6 model and period, for SSP1-2.6 (a), SSP2-4.5 (b), SSP3-7.0 (c) and SSP5-8.5 (d). A negative value shows the presence of a dipole of Z500 between Iceland and southwestern Europe. Boxplots are defined as in Fig. 3.

scenarios. Overall, the disparity between climate models makes it challenging to ascertain whether the identified pattern will increase or decrease in the future, irrespective of the scenario.







**Figure 7.** Temperature (TG15d) distribution of 1000 SWG simulations for four SSPs (a-d) and three climate periods (left to right) depending on the variable used for importance sampling (colours). The SWG simulations are obtained with the KACE-1-0-G model. Temperatures are adjusted by the median DJF temperature bias. Box plots are defined as in Fig. 3



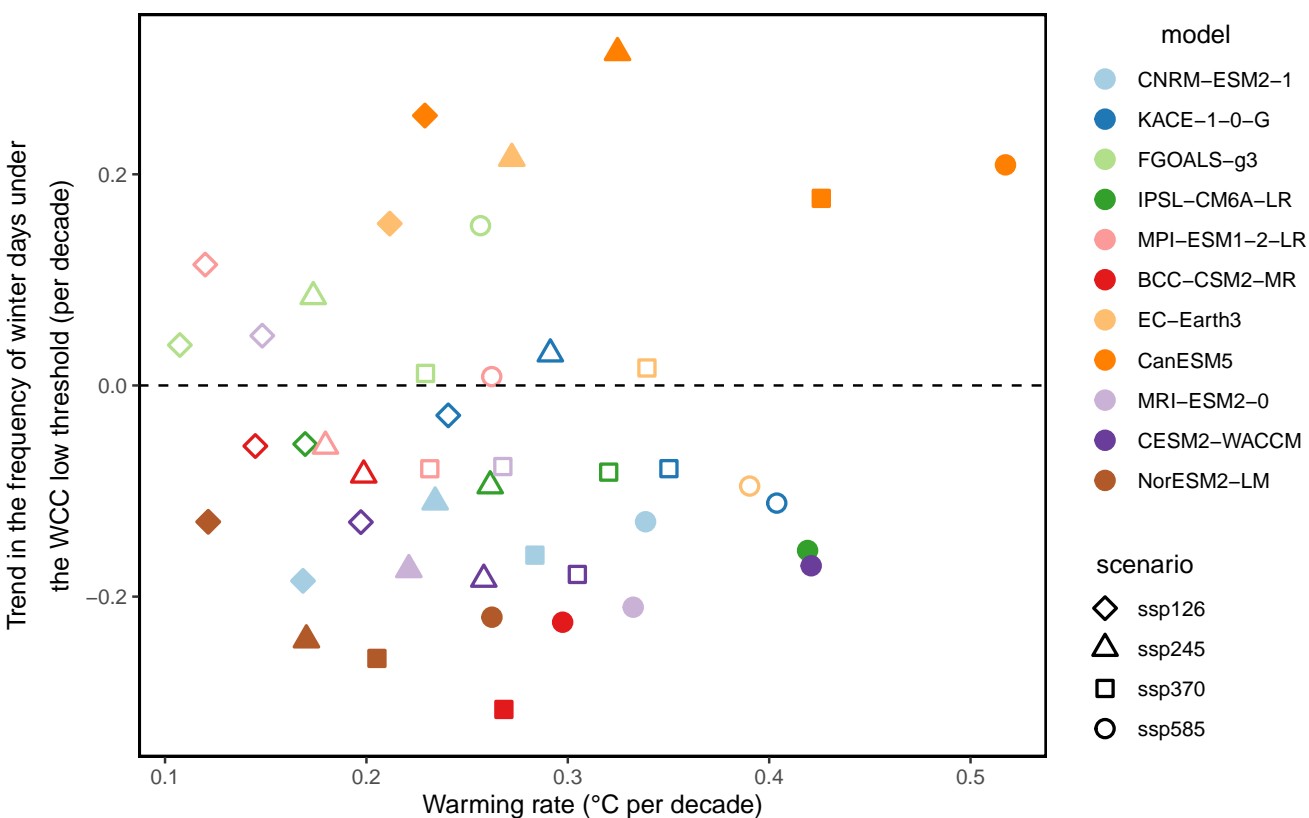

**Figure 8.** Yearly trend in the frequency of low WCC days by global warming rate between 1950 and 2100 for each 11 CMIP6 (colours) and scenario (shapes). Filled points outline the cases for which the WCC trend is significant according to a Mann-Kendall test.

## 5   Conclusions

In this study, we extended the findings of Cadiou and Yiou (2024) to 11 CMIP6 models to examine the evolution of extreme

cold spells in the future in France and to evaluate the ability of the models to reproduce the mechanisms leading to cold spells as observed in the ERA5 reanalysis.

Our study shows that cold spells of 15 days can still happen in France with moderate levels of global warming in the present and near future (i.e. at the 2050 horizon). This is coherent with previous works of Sippel et al. (2024) and Cadiou and Yiou (2024) which showed that extreme winter events like in 1962/63 could still happen in the present decades in Europe,

using ensemble boosting and ERA5-based SWG simulations. Therefore adaptation to a warming climate should not dismiss the possibility of low-likelihood high-impact cold spells (Cohen et al., 2020, 2023; Kim et al., 2014; Screen et al., 2018), in particular for transportation, health and energy sectors. However, our results indicate that, as expected, the intensity of very extreme cold spells decreases with global warming for all CMIP6 models. Cold extremes that occurred during the end of the $20^{th}$ century, such as winter 1962-1963 or February 1956, become almost impossible at the end of $21^{st}$ century for high levels



of warming. However ongoing electrification in Europe and the growing population may lead to increased vulnerability if extreme cold events were to occur, even if their occurrence is decreasing. With a high rate of electrification, France has an electric consumption that increases rapidly with low temperatures, compared to other European countries (RTE, 2023a). This could make France more vulnerable to very extreme cold spells, even under a reduced hazard.

    We also analyzed the changes in the intensity and dynamics of extreme cold events under different emission pathway sce-
narios. Consistently with previous results on heatwaves (Galfi and Lucarini, 2020; Noyelle et al., 2024), we find that the most extreme cold spells in ERA5 tend to have a similar atmospheric circulation with a high-pressure system over Iceland and a strong low-pressure system over southwestern Europe. When analysing the atmospheric circulation associated with the most extreme events in several CMIP6 models, we found that most models reproduce the same patterns of Z500 for cold spells, and that those patterns remain consistent across climate periods and scenarios. Some models display more variability in atmo-
spheric dynamics amongst simulations (FGOALS-g3, CanESM5) or a similar pattern but shifted towards Scandinavia or the South of Iceland (NorESM2-LM, CanESM5). But overall, all models tend to display a dipole of Z500 for extreme cold spells, which justifies the design of a circulation index for cold spells (WCC).

    We have demonstrated that the WCC circulation index is more adapted than the NAO index to predict low-temperature episodes over France. Conversely, the effect of this atmospheric pattern is singled out by nudging the SWG towards it without
direct constraint on temperature. We show that the dipole of Z500 identified from the coldest events in ERA5 remains a sufficient condition to trigger extreme cold spells, even in a warmer world or in models that reproduced the mechanisms less accurately.

    The SWG employed for the simulation of extreme cold spells yields some of the technical caveats previously highlighted by Cadiou and Yiou (2024) and Yiou and Jézéquel (2020). Our study evaluate how the mechanisms of cold spell are represented
in the selected CMIP6 GCMs, especially in a warmer climate. Our analysis of the WCC index indicates that most models do exhibit atmospheric patterns for cold spells that are comparable to those in ERA5 in the historical period. This WCC index is constructed for France, and could be adapted for other parts of Europe to simulate cold extremes and the associated atmospheric circulation.

*Data availability.* The ERA5 reanalysis and CMIP6 model data are publicly available, respectively at https://cds.climate.copernicus.eu/ and
https://esgf-node.ipsl.upmc.fr/projects/cmip6-ipsl/.

*Author contributions.* CC and PY conceived the experiments from the original code of PY. CC produced the numerical experiments and analyses. Both authors contributed to writing the manuscript.



*Competing interests.* The authors declare that they have no known competing financial interests or personal relationships that could have appeared to influence the work reported in this paper.

*Acknowledgements.* The authors acknowledge the support of the grant ANR-20-CE01-0008-01 (SAMPRACE: PY, CC). This work also received support from the European Union's Horizon 2020 research and innovation programme under grant agreement No. 101003469 (XAIDA: PY).





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
