# Peer review of "Intensity and dynamics of extreme cold spells of the 21st century in France from CMIP6 data"

_EGUsphere, 2024_

## Author Response (AR1)

**Reply to Review of "Intensity and dynamics of extreme cold spells of the 21st century in France from CMIP6 data" (Reviewer #1)**

We thank the reviewer for carefully reading our manuscript and for their constructive remarks. Our replies are in red (see pdf).

**General comment**

This paper investigates the evolution of 15-day cold extremes over France in 11 CMIP6 models, using a Stochastic Weather Generator (SWG) to increase the sample size and thereby allow conditioning on very extreme events. The authors argue that this provides something like a statistical equivalent of ensemble boosting, which is a method applied within a physical model. The methodology used here was introduced in a companion paper (Cadiou and Yiou, https://doi.org/10.5194/egusphere-2024-612) where it was applied to reanalysis data. However, I cannot ascertain the current status of that paper, which is unfortunate. I have to admit that I do not have a good feel for the meaning of the statistics being produced by this methodology, but since the same methodology is applied across different epochs, I understand that the differences between epochs are meaningful (i.e. we are perhaps more interested in precision than in accuracy here). The particular choice of region and cold-extreme length were motivated by implications for energy system demand, and the choice of CMIP6 models by the ready availability of the output data. That might all seem a bit arbitrary, but the subject of cold extremes has been rather neglected until recently (except for a reduction in cold extremes being a classical indicator of climate change), and I can't imagine that the results would change with the inclusion of more CMIP6 models. Thus I believe that this study will be of interest to the climate science community.

I personally find the most interesting result to concern the connection between the WCC circulation pattern (which is not one of the canonical North Atlantic circulation patterns) and 15-day cold extremes over France. If I understand correctly, the fact that conditioning on temperature or on WCC appears to yield pretty much the same cold extremes makes the WCC a unique predictor of those cold extremes (i.e. both necessary and sufficient). Moreover, the WCC pattern appears to be reasonably well simulated in the CMIP6 models, and to not exhibit a robust response to climate change up to the warming levels investigated. These findings support the value of storyline approaches for this phenomenon, and the use of CMIP6 models, both of which are useful things for researchers to know.

My only criticism of the study is that it does seem a bit perverse to throw away ensemble members and then try to increase the sample size statistically from the single remaining member! This data set would seem to provide a perfect opportunity for validation of the SWG, to see whether it can reconstruct the missing variability, since for some models there are presumably reasonably large ensembles available.

Thank you for this comment. The reason why we only used 1 run per model (and the first available in lexicographic order on the IPSL computing server, if runs are labelled as rXiYfZpU) is very prosaic: some studies in bias correction "just" use one GCM run (e.g.

François, B., Thao, S. & Vrac, M. Adjusting spatial dependence of climate model outputs with cycle-consistent adversarial networks. *Clim Dyn* **57**, 3323–3353 (2021). <a href="https://doi.org/10.1007/s00382-021-05869-8">https://doi.org/10.1007/s00382-021-05869-8</a>; Bohan Huang *et al* 2024 *Environ. Res. Lett.* **19** 094003, **DOI** 10.1088/1748-9326/ad66e6, just to mention a few). The r1i1p1f1 run also deserved more analyses from its conceivers (Boucher et al. (2020). Presentation and evaluation of the IPSL-CM6A-LR climate model. *Journal of Advances in Modeling Earth Systems*, 12, e2019MS002010. <a href="https://doi.org/10.1029/2019MS002010">https://doi.org/10.1029/2019MS002010</a>).

To tackle the issue raised by the referee, we checked the IPSL ensemble (33 members, with a convenient computer file format). The other models we consider in the paper have either small(er) available ensemble sizes or inconvenient file formats. The run we used in the manuscript (r1i1p1f1) seems to yield extreme cold spells that are warmer than in the 32 other simulations (blue points in Fig. 1 below). One member contains a 15-day event that is much colder than what has been observed (in ERA5), and colder than what is simulated with the SWG based on the r1i1p1f1 run (see Fig. 1 below). We now discuss this issue in the revised manuscript. In particular, we compare SWG simulations starting from r1i1f1p1 and r14i1f1p1 to the coldest TG15d event in r11i1f1p1 (see Appendix). Two new figures were added to the text (in Appendix A), and a discussion on the necessity of using large ensembles (rather than long simulations) to sample extremes is provided.

Figure 1: Temperature (after first order bias correction) of 5 coldest 15-day cold spells (TG15d) in each of the 33 IPSL run members. The blue dots on the left outline the coldest TG15d in the first run (r1i1p1f1), used in the study. The red dots are for the r14i1p1f1 run. The colored boxplots represent the empirical probability distributions of the SWG simulations starting from the coldest blue or red dots. The dashed line represents the "observed" value in 1987 in ERA5. The horizontal dotted line is the average of the yearly coldest TG15d value between 1950 and 1999 in the IPSL model runs (around 0°C). The average value of TG15d is around 4°C (i.e. outside of the figure).

**Minor comments**

My other comments are all minor:

Line 10: You should say what the outcome was of this analysis of the CMIP6 models.

Ok, we clarified the abstract about the results.

Line 57: Blackport and Screen (2020) is a curious citation to support this statement since it argues precisely the opposite.

Indeed. The sentence is corrected.

Lines 122-125: Was the K-S test applied to the raw model data or the bias-adjusted model data? The latter would be more informative, I think.

The K-S test was applied to the raw model data. This is clarified in the text. We will also perform a K-S test on the bias adjusted climate model time series, which allows keeping all considered models.

Lines 255-256: Is this after bias adjustment? That is what the caption to Figure 2 suggests. If so, that seems to be quite an interesting finding.

Yes, this is after bias adjustment. This is clarified in the text.

Figure 3: Can you clarify how the whiskers are constructed? The caption says the lower whiskers are constructed symmetrically to the upper whiskers, but they don't look symmetric in the figure. Also, in a few cases (notably IPSL for the middle epoch of SSP3-7.0) there would appear to be a very large number of outliers in the cold tail, which seems highly relevant for this study; what do you make of those?

The whiskers are constructed with an *algebraically* symmetric formula (i.e.  $min[max(T), q50 + 1.5 \times (q75 - q25)]$  for the upper whisker and  $max[min(T), q50 - 1.5 \times (q75 - q25)]$  for the lower whisker), which, depending of the distribution, does not necessarily result in symmetric boxplots. This is a standard formula for boxplots. With this formulation, non-Gaussian distributions can lead to many "outliers". This is mentioned in the caption of Fig. 3.

Lines 305-306: This is an interesting point, since it is quite common to look at the correlation between circulation indices and seasonal mean temperatures across the entire distribution. Can you say a bit more (rather than just a caveat, which is not so informative) about the extent to which the correlations in the tail that you are analysing are related, or not, to correlations across the entire distribution? That would be useful information for the reader, because it would guide any subsequent studies.

We now explain that correlation is generally obtained from the core of the probability distributions, and might not be very informative on the distribution tails. "Pushing on" the

WCC index (towards low values) is similar to the "do" action described by Hannart et al., 2016).

Lines 364-366: This statement is based on Figure 8, but I am having trouble reconciling Figure 8 with Figure 6. In Figure 6, I can hardly see any difference between the four panels for the third epoch, which is where one should presumably see the sensitivity to scenario.

Figure 6 (now Figure 4) displays the WCC index for SWG simulations, which are conditioned on extreme cold events. These figures show that the atmospheric configuration associated with extreme cold events is indeed little affected by the scenario and level of warming. Figure 8 shows the evolution in the frequency of a low WCC index in the daily circulation. Therefore, a model could yield a decrease in the frequency of low WCC days with warming, while still displaying a low WCC when an extreme cold spell occurs. This has been clarified in the text (L368).

Figure 7: It looks by eye as if the difference between the coldest events (those conditioned on either tas or WCC) and the rest increases as climate warms, which would seem to go against the general expectation of the coldest extremes over western Europe warming faster than the overall distribution (because of the reduced temperature difference with the Arctic). Can you comment on this? More generally, it would be useful to place this analysis within the context of the wider literature on projected changes in temperature extremes.

Figure 1 of Cadiou and Yiou (WCD, 2025) shows that the temperature trend of winter averages is similar to the trend of 3 day to 30 day averages over France (between 1950 and 2020). In our case, the "conditional" (to NAO, AR1...) SWG simulations do not reflect significant trend differences. For instance, the trends are detailed for SSP1-2.6 and KACE-1-0-G in the table below. The tas-SWG and WCC-SWG simulations exhibit a higher trend than the control (AR1) simulations, but not the simulations without importance sampling ("None"). Furthermore, the confidence intervals are very wide because there are only three periods. Therefore, we cannot conclude that there is a significant difference in warming between the mean and extremes.

| Importance sampling variable | Trend Coefficient (°C per 50-year period) | Lower bound | Upper bound |
|------------------------------|-------------------------------------------|-------------|-------------|
| None                         | 0.69                                      | -0.20       | 1.58        |
| tas                          | 0.80                                      | -2.95       | 4.54        |
| WCC                          | 0.65                                      | -3,88       | 5.18        |
| AR1                          | 0.17                                      | -1.09       | 1.44        |
| NAOi                         | 0.06                                      | -3.17       | 3.30        |

Table 1: Trend (in °C per 50-year period) in warming in the SWG simulations depending on the variable used for importance sampling across the three 50 periods in Figure 7.

Those results are similar (albeit not completely comparable to) what was obtained by Ribes et al. (BAMS, https://journals.ametsoc.org/view/journals/bams/aop/BAMS-D-24-

0013.1/BAMS-D-24-0013.1.xml), with a different methodology, based on extreme value theory. We added a reference to Röthlisberger and Papritz (2023), who analyzed the mechanisms leading to extreme cold temperatures (albeit for 1 day events).

Lines 654-656: This submitted version of Sippel et al. should be deleted since the published version is provided by the very next entry in the reference list.

Ok, this is corrected. We trust that the Copernicus editorial office will spot all those small problems.

**Reply to Review of "Intensity and dynamics of extreme cold spells of the 21st century in France from CMIP6 data" (Reviewer #2)**

We thank the reviewer for their careful reading of our manuscript and their constructive remarks. Our replies are in red.

**General comment**

Reviewer's comment on "Intensity and dynamics of extreme cold spells of the 21st century in France from CMIP6 data" by Cadiou and Yiou (2025)

The authors study historical, present and future extreme cold spells over France in ERA5 and CMIP6 models. They use a stochastic weather generator based on circulation analogues and importance sampling, which is a sort of stochastic rare event sampling algorithm. The paper concludes that the intensity of extreme cold spells decreases in the future as global warming progresses, however impactful cold spells may still occur in the near future and should not be overlooked. Furthermore, the paper evaluates the ability of CMIP6 models to realistically simulate the circulation anomalies leading to extreme cold in France.

The risk of future extreme cold spells is understudied and tends to become underestimated due to the focus on the increasing frequency and intensity of hot extremes as a consequence of global warming. This might exacerbate the general vulnerability of our society to cold extremes. Thus, the topic and the message of this paper is highly relevant. Furthermore, the paper is well written and clearly structured.

Thank you for your positive comment.

**Minor comments**

I suggest that the authors implement following minor corrections/changes:

1. The authors do not discuss the limitations of the methodology, but point instead to already published work. Since the rare event algorithm is the essential element of this study, I think that the authors should extend the paragraph about limitations in Sec. 5 and discuss, in a concise way, the main assumptions/limitations of the algorithm in this work as well. For example, this stochastic rare event sampling algorithm cannot generate new atmospheric states, but is based on a resampling of already explored atmospheric configurations. This and similar limitations should be mentioned and discussed.

Ok. the caveats of the SWG are more thoroughly discussed in section 5. One important issue was raised by Referee#1 (first comment). We provide an additional analysis to illustrate this GCM-dependent caveat.

2. This shall be an independently published work, thus I ask the authors to shortly summarise what they have done in the first paragraph of Sec. 5, instead of only referring to Cadiou and Yiou (2024).

Ok, the results of the previous paper Cadiou and Yiou (WCD, 2025) have been summarised.

3. It is confusing that Fig. 6 is discussed before Fig. 4 & 5. I suggest to reorder the figures: what is now Fig 6 should be shown before Fig 4 and Fig 5.

Ok, the Figures have been reordered to match the text.

4. L 136-137: Was the linear trend removed grid-point-wise?

Yes it was. This is clarified in the text

5. L 174-178: I don't understand how the content of this paragraph leads to the final statement: "In essence, we are evaluating ...". Some additional clarifications would be helpful.

We actually no longer do this (which was actually a key point of the paper of Cadiou and Yiou (WCD, 2025) who addressed a different question). This sentence is removed.

L 247-250: The paragraph on testing causal relations could be clearer. For example
instead of writing "here several atmospheric indices", please mention the actual
indices.

The paragraph is clarified by displaying the atmospheric indices used and explaining the "do" action of Hannart et al. (2016).

7. L 6: sentence is not clear, should be rephrased. Past events cannot re-occur, but events similar to past events can occur in the future.

Ok, this sentence is rephrased.

---

## Author Response (AR2)

**Reply to referee #1 (in blue)**

I am pleased to see that the authors responded to my suggestion to test whether the SWG applied to a single ensemble member can somehow replace the other ensemble members, and find that it cannot. (It seems a bit like bootstrap with replacement, which is widely used to produce confidence intervals in our community, yet is fundamentally limited since it has to work with the events that have occurred, and cannot create new events. Thus it will inevitably underestimate uncertainty.) This is an important caveat, and hopefully will encourage those analysing CMIP ensembles to include more than one ensemble member from each model in their analysis. Appendix A is a useful way to manage this issue within the context of the existing paper.

Again, we thank the referee for this suggestion.

I just spotted a couple of technical issues with Appendix A. On line 453, I believe that "r10" should be "r11".

This is corrected.

And on line 457, I believe that "r14" should be "r11".

No. This is actually r14, as explained slightly later in the text. We chose not to use the coldest event of r11, which is much colder than the coldest TG15d of all the other simulations, but a more typical cold TG15d (i.e. r14). This is explained around line 445. We do not reach as cold temperatures as in r11 with the SWG starting with r14 conditions (Figure A1).